# Barriers to implementation of domestic violence prevention policies and programs in northwestern Ethiopia: A qualitative implementation research

Agumasie Semahegn[1,2,3], Kwasi Torpey[2], Adom Manu[2*], Nega Assefa[1], Naana Akyiamaa Agyeman[2], Augustine Ankomah[2]

1 College of Health and Medical Sciences, Haramaya University, Harar, Ethiopia, 2 Department of Population, Family and Reproductive Health, University of Ghana School of Public Health, Legon, Accra, Ghana, 3 Centre for Innovative Drug Development and Therapeutic Trials for Africa (CDT-Africa), College of Health Sciences, Addis Ababa University, Addis Ababa, Ethiopia

* amanu@ug.edu.gh

## Abstract

Ethiopia is a signatory to various international conventions, regional charters, and protocols related to violence against women, yet many women suffer domestic violence. To date, very little is known about how these conventions and protocols are being implemented, and the barriers associated with their implementation. This study explored the barriers to implementation of domestic violence against women prevention policies and programs in northwestern Ethiopia. We conducted a qualitative study using in-depth interviews, key informant interviews, and focus group discussions among 43 participants. The study participants were purposively selected based on their key roles and positions in implementing policies and programs that aim to prevent domestic violence against women in the study area. The interviews and discussions were audio-recorded after obtaining consent from each study participant. Data were transcribed, coded, and thematically analyzed using NVivo 11 software. Implementation of domestic violence prevention policies and programs at the local level is fraught with many budgetary constraints, poor planning, non-adherence to planned activities, lack of political will and commitment in the local settings, competing priorities, poor program integration, and weak inter-sectoral collaboration. Therefore, future interventions that would sustain and synergize domestic violence prevention through the intersectoral collaboration of key actors, ensuring budgetary issues, improving local governors' will and commitment, and transforming deep-rooted inequitable gender -norms for successful domestic violence prevention policies and programs implementation.

## Background

Globally, one in three women experience at least one form of violence against women (VAW) in their lifetime [1–3]. The United Nations Declaration on the Elimination of Violence Against Women defines VAW as any act of gender-based violence that results in or is likely to

**Data availability statement:** All data are in the manuscript and/or Supporting Information files.

**Funding:** This work was supported through a PhD scholarship by the WHO/TDR, Special Program for Research and Training in Tropical Diseases, hosted at the WHO and co-sponsored by UNICEF, UNDP, the World Bank, and WHO, Grant No. B40300 to AS, through the University of Ghana. The scholarship was for capacity building. The funders had no role in study design, data collection and analysis, decision to publish, or preparation of the manuscript.

**Competing interests:** The authors have declared that no competing interests exist.

result in, physical, sexual, or psychological harm or suffering to women, including threats of such acts, coercion, or arbitrary deprivation of liberty, whether occurring in public or in private life [4]. It remains a huge threat to women's health and a significant public health burden [5]. Nevertheless, in many societies, domestic VAW is overlooked and considered a minor and private matter [5–7] despite the serious implications it has on victims, their families, health care costs, and productivity [8,9].

Domestic VAW is a manifestation of inequality and power imbalance in relationships [10]. In addition, domestic VAW is associated with critical reproductive health concerns such as low utilization of skilled birth attendance, reduced antenatal care, high unintended pregnancy, and unsafe abortion that result in poor reproductive health outcomes [1,4,5,10–18]. To offset the debilitating consequences of domestic VAW, the United Nations is aiming to create an enabling environment for women's rights, political, and economic empowerment, and legal protection by 2035 [19,20]. However, program integration, ownership, and scaling-up processes are very limited, and almost all of the gender-transformative health programs have relied on non-governmental organizations in low- and middle-income countries [21].

In Ethiopia, domestic VAW is still high compared to other low- and middle-income countries. The lifetime prevalence of VAW varies from one part of the country to another [22–24]. The differences in lifetime prevalence from different studies may be due to geographical and contextual differentials within the country [Ethiopia] [25], different methodologies, and the scope and coverage of these studies [14,26]. For example, the 2016 Ethiopian demographic and health survey reported that the prevalence of VAW ranged from 9% in the Somali regional state to 38% in Oromia [25]. Despite the prevalence variations, a recent systematic review and meta-analysis of 36 studies revealed a pooled lifetime prevalence of domestic VAW of 46.9% [26]).

Despite local and international human rights laws such as the Universal Declaration of Human Rights, Ethiopian women continue to suffer from violence and abuse [27]. In response to the high domestic VAW in Ethiopia, the government has endorsed various international and regional conventions, charters, and protocols, such as the Universal Declaration of Human Rights (1948) [27]; Conventions on the Elimination of all Forms of Discrimination Against Women, Article 1 [28]; the African Charter on Human and Peoples Rights, Article 3 (1-2) [29]; Beijing Platform for Action (1995); Committed to Safeguarding Women's Rights [30], and the Maputo Plan of Action (2016-2030) [20]. These documents give attention to the implementation of international, regional, and national legislation to create a conducive environment for women and girls. Additionally, the Sustainable Development Goal (SGD-5) aims at achieving gender equality and empower all women and girls. Specifically, SGD target 5.2 focuses on eliminating all forms of VAW and girls in public and private spheres, including trafficking and sexual and other types of exploitation. This makes it imperative for effective implementation of domestic violence prevention policies and programs at the grassroots level.

In addition, the government of Ethiopia has shown commitment by incorporating women's issues in the National Population Policy of 1993, which stipulates the minimum legal age of marriage as 18 years for both sexes. Setting the minimum age of marriage at 18 years among other things, aimed at reducing the high prevalence of child marriage in the country, and indirectly improving women's health and wellbeing [31]. Articles 14-18, 25, 34 (1-5), and 35 (1-9) of the 1994 Constitution of the Federal Democratic Republic of Ethiopia provide fundamental liberties to ensure gender equality and safeguard women's and human rights [32]. Again, the Criminal Code of Ethiopia under Proclamation No. 414/2004 ensures equality before the law (Article 4) and criminalizes any injury and suffering caused to women (Article 561) [33]. Furthermore, the Revised Family Code of Ethiopia specifies among other things conditions of marriage including equal rights of access to and control over resources (Article 42), respect

and support between partners (Article 49), and equal rights in the management of the family (Article 50) [34]. The Government of Ethiopia has launched a gender mainstreaming program in different sectors with an implementation manual to enforce existing policies [35], and standard operating procedures to respond to and prevent sexual VAW and girls. Although all these policies and legislations are in place, evidence shows that public awareness about their existence is very low including poor implementation [36].

Domestic VAW is a pervasive public health phenomenon in Ethiopia that seriously undermines women's and girls' health and social status. The most common forms of domestic violence include physical, sexual, and psychological. VAW is both a cause and effect of gender-based violence, child marriage [37–39], female genital mutilation [39,40], and other harmful traditional practices [39,41,42]. Nevertheless, the implementation of domestic VAW prevention and response programs is overlooked as a result of old-fashioned inequitable and patriarchal gender norms that give the wrong impression of domestic VAW as a minor and private matter. It is a curious contradiction that despite the various domestic VAW prevention policies and laws in the country, many women in Ethiopia continue to suffer domestic violence by their intimate partners. To date, very little evidence exists about the implementation of domestic VAW prevention policies and programs at the grassroots level. The few existing policies and legislation against domestic VAW in the country are poorly implemented. Therefore, this study sought to identify the barriers to poor implementation of domestic VAW prevention policies and programs in northwestern Ethiopia.

## Methods

### Ethics statement

The protocol was reviewed and approved by the Institutional Health Research Ethical Review Committee (IHRERC) of the College of Health and Medical Sciences at Haramaya University (*Ref. No. IHRERC/146/2017*). The study was conducted in accordance with the Helsinki Declaration. All the study participants were informed about the purpose of the study and were provided with the interview guide before the commencement of the interviews. A schedule of interviews was decided upon voluntarily. Verbal and written informed consent were obtained from participants in the current study. Informed consent was obtained from the legal guardian of illiterate participants for study participation. Confidentiality of the information was maintained by avoiding personal identifiers and password locking in computers with stored data.

### Study design

This qualitative study is part of a larger community-based quasi-experimental implementation research conducted in the Awi Zone, northwestern Ethiopia, from November 15, 2017, to November 15, 2018 [43]. The data presented in this manuscript relates to the views of individuals connected with the implementation of existing policies and initiatives in Ethiopia that seek to prevent domestic VAW. The study was guided by the Consolidated Framework for Implementation Research (CFIR) [44,45] and content analysis. The CFIR framework, among other things, assesses implementation barriers based on five established domains: namely evidence-based intervention (implementation of existing policy and legislations), inner setting, outer setting, individual behaviors, and process-related factors. Also, the CFIR framework looks at evidence gaps, which shows that many theories have focused on understanding individual behavior changes. Nevertheless, little research has been done to understand the dynamic relationship between individual behaviors, organizational settings, and overall environment. Furthermore, frequently cited social, cognitive, and planned behavior

theories mainly focus on the intention and behavior of individuals. The CFIR was used as a pragmatic meta-theoretical framework to assess, inner and outer settings, individual characteristics (behavior), and implementation process comprehensively [44,46]. Domestic VAW is a common phenomenon in Ethiopia [22–24]. This study utilized the Consolidated Criteria for Reporting Qualitative Studies (COREQ) framework [47] to ensure comprehensive reporting of the methods and results (see the details in S1 Checklist).

## Study setting

The study was conducted in Awi Zone in the Amhara Regional State, northwestern Ethiopia. Injibara town is the administrative center, located 447 kilometers from Addis Ababa in northwestern parts of Ethiopia. According to the Awi Zonal Health Department report, Awi Zone had a population of 1,285,242, in 2018, of which 50.9% are women [48]. Evidence from the national census of Ethiopia conducted by the Central Statistical Agency (CSA) indicates that 87% of the population lives in rural areas. It has 215,564 households with an average of 5 persons per household. About 93.5% of the population are Ethiopian Orthodox Christians, and 5.4% are Muslims [49].

## Sampling method

The principal investigator first consulted local leaders to identify key officials and individuals whose job roles centered around gender-based violence prevention and gender policy implementation and enforcement. Forty-three participants were recruited using the purposive sampling technique based on their respective gender-related roles and positions in the study area. The sample recruitment was guided by information saturation or redundancy of data from study participants. Thereafter, the study participants comprising Cluster Police Officers, Local Leaders, Government officials including women, Community Health Extension Workers, community representatives (called health development armies) in the communities, and religious leaders were recruited as participants for in-depth interviews (IDIs), key-informant interviews (KIIs) and focus group discussions (FGDs). The study participants who had positions related to gender at targeted government offices and/or departments, such as Health, Women and Child Affairs, and Justice and Security were purposively recruited for the KIIs. Lastly, different focus group discussions for men and women were conducted using discussants from implementors of VAW prevention programs. Brief explanations about the main objective of the study were given to the local leaders and the study was participants by the principal investigator and data collectors; an overview of the interview guide was given to the study participants, and a schedule for interviews was booked.

## Data collection procedures

Investigators and trained interviewers collected data face-to-face using IDIs, KIIs, and FGDs methods to explore barriers to the existing policy implementation to prevent domestic VAW. We used semi-structured interview guides to moderate the interviews that were developed through intensive literature review, consulting researchers, and reviewed and validated by senior experts in the field. IDIs explored individual perceptions about intimate partner violence, gender equality, and attitudes towards justified wife-beating, while KII explored the overall situation of VAW and gender equality in the locality, as well as the prevention measures, and policy implementation barriers. Additionally, the KIIs explored the availability of collaborators and program integration possibilities, community attitude towards domestic VAW, and gender equality. Both IDIs and KIIs lasted 30-45 minutes. In addition, each FGD comprised of 8-10 discussants was conducted using a guide to assess the general community

attitudes, perception towards domestic VAW, level of implementation of domestic violence preventive measures, and implementation bottlenecks. To ensure FGD discussants, especially women had the freedom and the confidence to express their views, separate FGDs were conducted for two women's and two men's groups and were facilitated by three members of the research team (moderator and two note-takers). KIIs were conducted in offices with prior appointments with participants for convenient time and to ensure privacy; while IDIs and FGDs were performed in homes and villages, respectively. Discussions were audio-recorded while note takers captured nonverbal communication cues (physical gestures) during the discussions. The FGDs lasted between 60-90 minutes.

## Reflexivity statement

The two male authors (AS, NA) who hold PhD degrees with extensive experience in field research facilitated the study and conducted the interviews. These individuals grew up in the rural context of Ethiopia where domestic VAW is high and considered a common practice in marital life and are familiar with VAW issues. The lead author (AS) has extensive training in gender issues and is a key advocate for balanced gender equity and equality to prevent VAW. This issue motivated him to start his research career on domestic VAW in Ethiopia and its associated factors. He has extensive research experience on VAW and has published widely in this area. He is aware of the basic principles of VAW research and ethical considerations. In addition to understanding the real-world context of the study area, he has conducted a rigorous systematic review of VAW to understand the extent of VAW, and factors associated with VAW and used the relevant found for tool development. He conducted a reliability estimate analysis for the adapted tool for the formative assessment of the VAW study, and this qualitative study is a part of that quasi-experimental study. Although he had been involved in qualitative data collection, coding, and analysis, he was very careful to minimize insider biases by using interview guides and research assistants. All the research processes were regularly reviewed and validated by senior co-authors and stakeholder representatives who have no connection with the study site and the prominent culture of the society. All authors had been working in higher education institutions in Ethiopia and Ghana, as faculty.

## Data management and analysis

Data were transcribed and translated verbatim from the local languages (Amharic and Awigni) into English by playing and re-playing the audio tapes and referring to the summary notes taken during the interviews or discussions. The study participants' socio-demographic data were entered into SPSS (version 23.0) for the descriptive analysis (Table 1). The transcripts were thoroughly reviewed and imported into NVivo 11 [50] for coding and thematic analysis [51,52] by two experts. Each expert independently read the transcripts thoroughly to familiarize themselves with the data, discerned patterns using predetermined codes, and generated new codes as necessary. The generated codes were compared, refined, and finalized. Overall, our analytical framework employed a hybrid approach of deductive and inductive coding [53,54]. The deductive approach was based on the consolidated framework, and specific codes were generated, while we inductively identified additional codes based on the content of the data to reflect participants' views. The transcribed data from interviews and FGDs were arranged into thematic areas. Similar ideas were read and re-read to better understand the world of the participants. Thus, emerging themes and sub-themes were identified accordingly to form the basis of our study findings. Participants' narrative quotes were presented using participants' anonymous codes Additionally, parent and child nodes corresponding to themes and sub-themes were constructed according to the flow of information

**Table 1. Background characteristics of study participants.**

| Variables | | Frequency n=43 | Percent |
|---|---|---|---|
| **Age** (years) | Mean (±SD) | 37.5 (±8.6) | |
| | ≤36 | 22 | 51.2 |
| | >36 | 21 | 48.8 |
| **Sex of respondent** | Female | 25 | 58.1 |
| | Male | 18 | 41.9 |
| **Occupational status** | Housewife | 8 | 18.6 |
| | Farmer | 6 | 14 |
| | Merchant | 3 | 7.0 |
| | Government employee | 24 | 55.8 |
| | Others | 2 | 4.6 |
| **Educational status** | Illiterate | 2 | 4.7 |
| | Able to read and write | 2 | 4.7 |
| | 1–6 Grades | 4 | 9.3 |
| | 7–12 Grades | 11 | 25.6 |
| | 12+ | 24 | 55.8 |
| **Organization represented** | Women and Child Affairs Department | 7 | 16.3 |
| | Police and Security Department | 6 | 14.0 |
| | Health Department | 16 | 37.2 |
| | General Community Representatives | 13 | 30.2 |
| | Justice Department | 1 | 2.3 |
| **Residence** | Urban | 23 | 53.5 |
| | Rural | 20 | 46.5 |

about participants' perspectives. Finally, the implementation barriers were summarized using the CFIR model [44,46].

## Results

A total of 43 participants were involved in the study. Of these, 34 participated in FGDs, five were involved in KIIs, and four partook in IDIs. The mean age of the study participants was 37.5(±8.6) years, and more than half (55.8%, n=24) were government employees (Table 1).

### Barriers to community-based intervention implementation

The operationalization of existing policy documents and research evidence recommendations to prevent domestic VAW have been affected by various barriers. Participants mentioned various barriers at different levels that influenced the implementation of existing policy and legislation, including financial constraints, lack of local political will and commitment due to competing priorities, lack of awareness about existing policies and programs, poor planning, low involvement of key stakeholders, limited collaborations and integration, and existence of entrenched and skewed community traditional gender-norms.

### Financial constraints

Financial constraints were one of the most frequently cited barriers to domestic VAW prevention policy implementation. Participants explained that the government has a budgetary policy that mandates every sector to allocate two percent of their annual budget for

gender-related activities, but this is hardly followed, as the allotted budget is not used for the intended purposes.

One female discussant in the FGDs said

> "…*no special budget is allocated for gender equality or domestic violence prevention in our setting…during annual budget allocation in our administration, they address the issue of women and children affairs after considering other sectors' budget allocations*"

(F, FGD$_{p1}$).

## Lack of political will and commitment

Lack of political will and commitment (mainly local politicians) appeared as one of the barriers to implementing existing policy to prevent VAW in the study area. Most of the study participants agreed that stakeholders have competing priorities to the detriment of domestic VAW prevention programs. In addition, the presence of several committees with many tasks caused difficulty in the implementation of domestic VAW prevention programs. Participants indicated that domestic VAW prevention issues are just recorded in policy documents, but very limited practical attention is given to it in most settings. Practical intervention to address domestic VAW issues is not being properly implemented. In addition, participants were of the view that gender mainstreaming has not been actively implemented because of the apparent indifference of many stakeholders, lack of commitment and attention from local politicians, poor involvement of the implementers, and lack of full engagement within the community. These barriers affected the implementation of existing policies and programs to prevent domestic VAW in the study area.

One male FGD participant remarked

> "…*to ensure gender equality, politicians should teach and serve as role models. They should start from their offices and homes. Sadly, they are not good role models. They are not also working but only talking and putting the agenda on the paper… 'only paper value'… Generally, implementation of policies at the local level is so poor…*" (M, FGD$_{p1}$)

The lack of political will and commitment is manifested in apathy among officials to consciously develop appropriate and well-targeted plans for domestic VAW prevention activities. One female FGD participant commented, thus:

> "…*they (top officials) do not consider gender equality and domestic VAW prevention during zonal administrative planning. So, we do not have a plan to do this type of training to reduce domestic VAW, and we did not do anything based on plan…*" (F, FGDp$_6$)

Furthermore, participants also mentioned that the absence of an independent institution responsible for program implementation affects awareness creation and undermines effective domestic VAW prevention efforts. One of the female discussants in an FGD said:

> "…*Officials have overwhelmingly busy schedules, and lack of independent responsible body (institution) to coordinate this (domestic violence prevention issue) is a problem because Women and Children Affairs Office is almost not functioning well, they are only implementing politics, very superficially. No one is asking the women their life experiences in their home…*" (F, FGD$_{P3}$)

In addition, one of the key informants complained that poor grassroots engagement due to the absence of a responsible agency to coordinate programs affects the effectiveness of implementation efforts. She indicated:

"…the *Women's and Child Affairs Offices has no structure at the bottom (sub-district level) to engage the community very well…*" (F, KII$_{002}$)

## Poor collaboration and integration with the existing program

Evidence from participants' views indicated that the existing policies and evidence-based interventions on domestic violence prevention are not being implemented due to a lack of inter-sectoral collaboration. Other participants mentioned that there is poor integration of domestic violence prevention activities into existing programs, such as the community health extension program and others. The following quotes from some of the participants illustrate poor collaboration and integration of programs:

"…*if we work in collaboration with the community health extension workers, implementation of programs for the prevention of VAW will be an easy task. It will not be an extra burden for us, and it will bring about the needed change…*" (M, FGD$_{adv1}$)

"**…**there is a huge gap in integrating efforts to prevent VAW. Perhaps, one sector may take the responsibility of leading a particular initiative, but may not be well integrated with other sectors to address VAW at the grass root level…*" (M, KII$_{001}$)

To throw more light on poor inter-agency collaboration, and the potential impact of such partnerships on domestic VAW prevention, one female FGD discussant said:

"…*domestic VAW prevention is not well integrated with the existing community health extension program. Integration is possible to enable implementation of domestic VAW prevention interventions…*" (F, FGD$_{p6}$)

The same participants also said,

"…*no one has been supporting us so far…we could not find any sector working on prevention of VAW to collaborate with or integrate our plan with…everybody speaks about program integration…but there is very little practical implementation to prevent domestic VAW…everything is pushed to the community health extension workers without adequate support…*" (F, FGD$_{p6}$)

Furthermore, a female key informant confirmed that the health extension program is firmly grounded in the communities and collaborating with them will be the best way to prevent domestic VAW, yet such integration does not exist.,

"…*although we have not been working that much with good collaboration, the health sector has a community-based health education program, which has been delivered by community health extension workers. If sectors could integrate it would not be difficult to tackle domestic violence. The health education program itself is a good enabler if we use it for the future. The health sector is an important implementer, better than other sectors. The community health extension workers are the leader of the 'women development army'…*" (F, KII$_{009}$)

## Traditional gender norms

Almost all participants agreed that the prevailing attitude toward traditional gender norms in the community serves as a major barrier to effectively implementing existing policies and interventions to

promote gender equality and prevention of domestic VAW. The community has an ingrained culture that supports traditional gender norms, for instance, there is a lack of positive attitudes towards gender equality and an absence of male involvement in domestic violence prevention activities. It was evident that community members and other stakeholders, such as politicians, local leaders, religious leaders, elders, and others, all lacked information and understanding of the negative effect of domestic VAW, and gender inequality norms were the barriers to the implementation of domestic VAW prevention interventions. For instance, most people in the community as well as other stakeholders perceive wife-beating (intimate partner violence) as a normal practice. One of a male FGD participant remarked:

> "…to ensure gender equality, politicians should be taught to be role models. They should start from their offices and homes. But so far, they have not been a good example to others. They have not also demonstrated deed but only talking 'paper value'…" (M, FGD$_{adv1}$)

The barriers to the implementation of domestic VAW are presented using the CFIR [46] (Fig 1).

## Discussion

This study explored the existing policy and program implementation status, aimed at preventing domestic VAW and its barriers in northwestern Ethiopia. Implementation of existing policies and programs related to gender equality and domestic VAW at the community level is influenced by budget constraints, lack of political will and commitment, poor integration, and collaborative works, presence of deep-rooted inequity norms, and stakeholders having competing priorities.

The implementation of the existing government policies is crucial to respond to and prevent domestic VAW. This study found that there is a gap in the effective implementation of existing policies about gender equality and domestic VAW prevention and control. This study's finding is consistent with the barrier reported by the WHO's [55] HIV/AIDS prevention policy through the integration of promoting gender equality and equity, and a systematic review of epidemiological studies [56]. To respond to this, necessary legal, regulatory, and policy reforms must be developed and implemented to strengthen service integration and

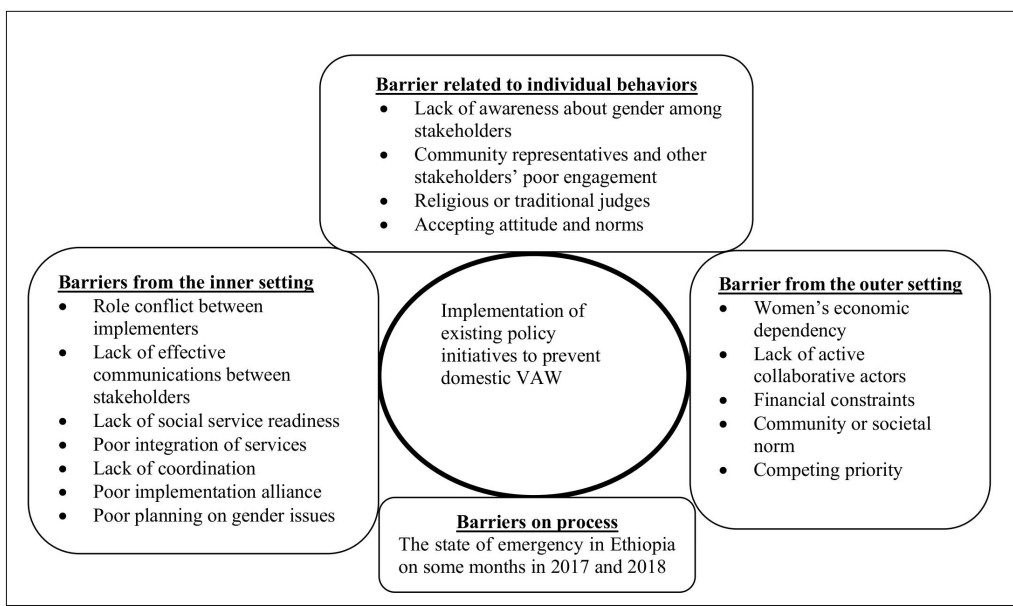

**Fig 1. Summary of the barriers to domestic VAW prevention program implementation.**

linkages to improve the efficiency of community-based interventions. Furthermore, it was found that the translation of research evidence into routine practice was hindered by a lack of commitment from local politicians, financial constraints, lack of stakeholder engagement and collaboration, poor planning and non-adherence to the plan, lack of awareness about domestic VAW among politicians, or implementation alliance and competing priorities.

In addition, this qualitative study identified implementation barriers consistent with other available evidence [57] revealing that government needs to address the political, social, and economic structures that subordinate women. Implementing national plans and making budget commitments to invest in actions by multiple sectors, including community and other stakeholders involving women and men can shift discriminatory social norms. Likewise, education and empowerment of women are fundamental. Furthermore, training health workers to identify and support survivors is a strategy to prevent and respond to violence [57]. A weak intersectoral response to domestic VAW in the health system has been a challenge. Strengthening the role of the health system in addressing domestic VAW [58,59] needs a comprehensive health-system approach as a serious health priority that helps healthcare providers to identify and support women subjected to domestic VAW through capacity building, and effective coordination between stakeholders.

Evidence has shown that community normative behavior towards domestic VAW has been improving. Community norm that accepts domestic VAW in Ethiopia is not an uncommon phenomenon [25,60,61]. Therefore, improving women's participation in decision-making and gender equitable norms makes a great contribution to reducing negative attitudes toward women and domestic VAW [62]. Likewise, to bring impactful changes to individuals and society at large, local resources, health services public health schemes, and communities should be mobilized. Intersectoral collaboration and integration are crucial to addressing the root cause of domestic violence, which is a global public health epidemic [63]. The presence of appropriate community structures, community health care, social organizations ('Idir' and 'Equib'), and strong political will from the government, are key for successful program delivery and effectiveness. Additionally, school-based gender clubs and non-governmental organizations working on domestic violence prevention and related issues, and responsiveness of community members are some of the great enabling opportunities that can facilitate the implementation of programs in promoting gender equality and prevent domestic VAW [64].

Intersectoral collaboration has to be improved as strategies across the social ecology (interacting, social, institutional, cultural, and political contexts) to achieve meaningful changes within the existing social and political structures. It is important to develop program components that are comprehensive and mutually reinforcing through collaboration and coordination instead of stand-alone interventions. In a multisectoral approach, as shown by this study's outcomes, changes in attitudes and behaviors may not need a generation but can be achieved within shorter timeframes if intervention models adhere to key principles for effective prevention of domestic VAW and girls. Women's movements have led to advocacy and action against domestic VAW and remain central in the design and implementation of high-quality programs to prevent domestic VAW. Based on evidence and promising practical models, greater investments are needed in programmatic innovations, research-activist collaborations, and health sector leadership to build even greater momentum for the primary prevention of domestic VAW and girls [59].

## Strengths and weaknesses of the study

Community-based research is very crucial to identifying problems in real-world settings and is appropriate for the generalization of the findings for better insight and to inform policy and programs. This study explored the implementation barriers to domestic violence prevention,

using mixed-method qualitative research that enhances the transferability of the findings. In addition, the study was conducted for a year to identify and assess the bottlenecks as a part of a pilot quasi-experimental interventional study (41). The individuals' experiences of other forms of domestic abuse and personal life stories were not explored due to the sensitive nature of VAW. Again, the focus of the present research was on barriers to the implementation of existing policies to prevent domestic VAW in Ethiopia. Thus, future studies need to explore these limitations to provide a better understanding of VAW in Ethiopia.

## Conclusions

The existing policy framework is a good reflection of the government's commitment and willingness to safeguard the rights of women and girls against domestic violence. Nevertheless, budgetary constraints, lack of commitment, poor integration and collaborative works, deep-rooted inequity norms, and stakeholders having competing priorities are critical barriers to successful domestic violence prevention policies and programs. Creating a synergistic environment for integration of domestic violence prevention program implementation with well-grounded existing programs will be extremely helpful in reducing domestic violence in northwestern Ethiopia.

## Supporting information

**S1 Checklist.** COREQ Reporting Checklist.
(DOCX)

## Acknowledgments

We would like to thank the School of Public Health, the University of Ghana, the World Health Organization (WHO) special program of Tropical Disease Research (TDR), and Haramaya University (Ethiopia) for their overall support.

## Author contributions

**Conceptualization:** Agumasie Semahegn, Kwasi Torpey, Adom Manu, Nega Assefa, Naana Akyiamaa Agyeman, Augustine Ankomah.

**Data curation:** Agumasie Semahegn.

**Formal analysis:** Agumasie Semahegn.

**Investigation:** Agumasie Semahegn.

**Methodology:** Agumasie Semahegn, Kwasi Torpey, Adom Manu, Nega Assefa, Naana Akyiamaa Agyeman, Augustine Ankomah.

**Project administration:** Agumasie Semahegn.

**Supervision:** Kwasi Torpey, Adom Manu, Nega Assefa, Naana Akyiamaa Agyeman, Augustine Ankomah.

**Writing – original draft:** Agumasie Semahegn.

**Writing – review & editing:** Kwasi Torpey, Adom Manu, Nega Assefa, Naana Akyiamaa Agyeman, Augustine Ankomah.

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
