## [Decision Letter · Decision Letter 0]

23 Aug 2024

PGPH-D-24-00970

Barriers to implementation of domestic violence prevention policies and programs in northwestern Ethiopia: A qualitative implementation research

Dear Dr. Manu,

Thank you for submitting your manuscript to PLOS Global Public Health. After careful consideration, we feel that it has merit but does not fully meet PLOS Global Public Health’s publication criteria as it currently stands. Therefore, we invite you to submit a revised version of the manuscript that addresses the points raised during the review process.

Please see the comments of one reviewer below, whose comments we now invite you to address. Please note that we have only been able to secure a single reviewer to assess your manuscript. We are issuing a decision on your manuscript at this point to prevent further delays in the evaluation of your manuscript. Please be aware that the editor who handles your revised manuscript might find it necessary to invite additional reviewers to assess this work once the revised manuscript is submitted. However, we will aim to proceed on the basis of this single review if possible.  

We look forward to receiving your revised manuscript.

Kind regards,

Hanna Landenmark

Staff Editor

Journal Requirements:

1. We ask that a manuscript source file is provided at Revision. Please upload your manuscript file as a .doc, .docx, .rtf or .tex.

2. We note that your Data Availability Statement is currently as follows: This was a qualitative study. Relevant data that support the findings of this study are presented within the content of the manuscript.

Additional Editor Comments (if provided):

Reviewers' comments:

Reviewer's Responses to Questions

**Comments to the Author**

1. Does this manuscript meet PLOS Global Public Health’s publication criteria ? Is the manuscript technically sound, and do the data support the conclusions? The manuscript must describe methodologically and ethically rigorous research with conclusions that are appropriately drawn based on the data presented.

Reviewer #1: Yes

2. Has the statistical analysis been performed appropriately and rigorously?

Reviewer #1: N/A

3. Have the authors made all data underlying the findings in their manuscript fully available (please refer to the Data Availability Statement at the start of the manuscript PDF file)?

Reviewer #1: Yes

4. Is the manuscript presented in an intelligible fashion and written in standard English?

Reviewer #1: Yes

5. Review Comments to the Author

Reviewer #1: The study employs a qualitative approach to investigate the challenges and barriers in implementing domestic violence prevention policies and programs in northwestern Ethiopia. Using the Consolidated Framework for Implementation Research (CFIR), the researchers conducted in-depth interviews, key informant interviews, and focus group discussions with purposively selected participants involved in policy implementation. The study utilized an established qualitative data analysis method, including thematic analysis using NVivo 11 software. The findings reveal multiple barriers across various domains, including evidence-based intervention, inner setting, outer setting, individual behaviors, and process-related factors. The results are presented clearly, supported by direct quotes from participants, and organized under thematic subheadings. This research provides valuable insights into the complexities of implementing domestic violence prevention strategies in the region and can inform future efforts to improve policy implementation and program effectiveness. Overall, the manuscript is technically sound.

I will present my comments by sections.

1. Introduction:

The introduction provides a comprehensive background on the prevalence and impact of violence against women (VAW) i) globally, ii) in Ethiopia and iii) the legal and policy framework in Ethiopia. It highlights the significance by citing statistics and referencing international declarations and local laws aimed at combating VAW, thus establishing the relevance of the study in the context of public health and human rights.

1.1. Line 56: "Domestic VAW is a manifestation of inequality and power imbalance in relationship" should be "in relationships."

1.2. Line 66: Lifetime prevalence of VAW varies from one part of the country to another – Varies how much? Any references?

1.3. Line 88: “Setting the minimum age of marriage at 18 year among” it should be “years”.

1.4. Line 104: "…. in Ethiopia that seriously undermines women and girls’ health and social status" should be "women's and girls’ health and social status.”

2. Methods

The methods section is appropriate for the study's objectives. It provides a clear and detailed description of the study design, setting, sampling method, data collection procedures, data management and analysis, and ethical considerations. The use of the Consolidated Framework for Implementation Research (CFIR) is well-justified and aligns with the study's aim to identify barriers to the implementation of domestic violence prevention policies and programs.

A few suggestions:

2.1. The methods section does not discuss the researchers' potential biases or reflexivity, which is important in qualitative research to acknowledge and mitigate any influence on data collection and analysis.

2.2. While the section mentions the use of IDIs, KIIs, and FGDs, it could benefit from more specific details on the interview guides' content and how they were developed.

2.3. Sample size. Why 43 participants? I didn´t find any justification for this number

2.4. What about data saturation? The section does not mention whether data saturation was achieved.

2.5. There is no information on the role of researchers during data collection and analysis, which could provide insights into how data were interpreted and any potential biases. Maybe add a section discussing the researchers' potential biases and how they were addressed during the study to enhance the credibility of the findings? Author´s contribution state that only one author conducted the data analysis, which it is not the usual protocol of qualitative data analysis. A single coder can have several implications in the analysis such as, lack of inter-coder reliability and reflexivity (For more information see, for e.g., COREQ protocol). Thus, confirmation bias should be acknowledged as a limitation of the study.

2.6. Provide more specific information on the content of the interview guides and how they were developed to give readers a clearer understanding of the data collection process.

2.7. Line 187: Was it then an “hybrid approach” to thematic analysis?

3. Results

The results section is appropriate for the study's objectives. It provides a clear and detailed presentation of the findings, including both quantitative demographics and qualitative data.

3.1 There is some repetition in the quotes and points made, particularly regarding the lack of political will and poor collaboration. Reduce repetition in the quotes and points made to streamline the content and improve clarity. For instance, political will and political commitment are closely related concepts, it would be useful to have a table with the description, focus and nature of the themes included in the results section.

3.2 The results present the KII results and FGDs in a mixed analysis, it would bring some clarity if the FGDs were classified in the citations such as, FGD1; FGD2, etc. It clearly help attribute quotes and findings to their respective sources and see which FGDs were more cited, for instance.

3.3 Line 288: punctuation

4. Discussion

The discussion provides a thorough examination of the barriers to implementing domestic violence against women (VAW) prevention policies in northwestern Ethiopia.

4.1. While barriers are well-identified, the discussion could benefit from more in-depth exploration of potential solutions.

4.2 The discussion relies heavily on qualitative findings. Incorporating some quantitative data could strengthen the arguments.

4.3. Some ideas, such as the need for intersectoral collaboration, are repeated multiple times without adding new insights. E.g. intersectoral collaboration (IC) was mentioned in sentences starting at line 348, 360, 372, 380. To avoid repetition and enhance clarity, this point can be consolidated into the same section. Or, if the goal of the authors is to indeed focus the discussion on IC, maybe divide it by parts such as, Role and importance of IC; Barriers to effective IC; Strategies for Improving IC. (This is clearly just a suggestion for authors consideration)

5. Strenghts and Limitations

5.1. Line 395-396: "This study used mixed qualitative methods to explore implementation barriers that are recommended for the findings' transferability."

• This sentence is a bit awkward. It could be rephrased for clarity, though it's not strictly a typo.

6. Conclusion:

6.1 Line 410: "...will be super helpful in reducing domestic violence in northeastern Ethiopia." instead of super, perhaps use "highly" or "extremely", a bit more formal.

• Northeastern or Northwestern?!

6. PLOS authors have the option to publish the peer review history of their article (what does this mean? ). If published, this will include your full peer review and any attached files.

**Do you want your identity to be public for this peer review?** For information about this choice, including consent withdrawal, please see our Privacy Policy .

Reviewer #1: **Yes: ** Liliana de Abreu

---

## [Decision Letter · Decision Letter 1]

19 Nov 2024

PGPH-D-24-00970R1

Barriers to implementation of domestic violence prevention policies and programs in northwestern Ethiopia: A qualitative implementation research

Dear Dr. Manu,

Thank you for submitting your manuscript to PLOS Global Public Health. After careful consideration, we feel that it has merit but does not fully meet PLOS Global Public Health’s publication criteria as it currently stands. Therefore, we invite you to submit a revised version of the manuscript that addresses the points raised during the review process.

I am willing to accept this manuscript for publication pending a few minor revisions to improve the clarity and readability of the text. I invite you to submit a revised version of the manuscript that addresses the points raised below.

Please use the COREQ checklist, or another relevant reporting checklist, to ensure complete reporting of the methods and results. A supplementary document should be included detailing where in the manuscript the checklist items are reported.I recommend using a language editing service to identify and correct a few remaining spelling and grammatical errors (e.g., inappropriate uses of the comma in lines 47, 51, 55; "resulting" in line 57 should be "result"; etc.).

We look forward to receiving your revised manuscript.

Kind regards,

Marilyn Naana Ahun, PhD

Academic Editor

Journal Requirements:

Additional Editor Comments (if provided):

Reviewers' comments:

Reviewer's Responses to Questions

**Comments to the Author**

1. If the authors have adequately addressed your comments raised in a previous round of review and you feel that this manuscript is now acceptable for publication, you may indicate that here to bypass the “Comments to the Author” section, enter your conflict of interest statement in the “Confidential to Editor” section, and submit your "Accept" recommendation.

Reviewer #1: All comments have been addressed

2. Does this manuscript meet PLOS Global Public Health’s publication criteria ? Is the manuscript technically sound, and do the data support the conclusions? The manuscript must describe methodologically and ethically rigorous research with conclusions that are appropriately drawn based on the data presented.

Reviewer #1: Yes

3. Has the statistical analysis been performed appropriately and rigorously?

Reviewer #1: N/A

4. Have the authors made all data underlying the findings in their manuscript fully available (please refer to the Data Availability Statement at the start of the manuscript PDF file)?

Reviewer #1: Yes

5. Is the manuscript presented in an intelligible fashion and written in standard English?

Reviewer #1: Yes

6. Review Comments to the Author

Reviewer #1: Thank you for carefully addressing the comments.

7. PLOS authors have the option to publish the peer review history of their article (what does this mean? ). If published, this will include your full peer review and any attached files.

**Do you want your identity to be public for this peer review?** For information about this choice, including consent withdrawal, please see our Privacy Policy .

Reviewer #1: **Yes: ** Liliana Abreu

---

## [Decision Letter · Decision Letter 2]

7 Jan 2025

PGPH-D-24-00970R2

Barriers to implementation of domestic violence prevention policies and programs in northwestern Ethiopia: A qualitative implementation research

Dear Dr. Manu,

Thank you for submitting your manuscript to PLOS Global Public Health. After careful consideration, we feel that it has merit but does not fully meet PLOS Global Public Health’s publication criteria as it currently stands. Therefore, we invite you to submit a revised version of the manuscript that addresses the points raised during the review process.

Thank you for revising your manuscript and for providing a completed COREQ checklist in your response to the reviewers. Before we can proceed with your submission please upload the completed COREQ reporting checklist as a supplementary file alongside your revised manuscript.

We look forward to receiving your revised manuscript.

Kind regards,

Emma Campbell, Ph.D

Staff Editor

Journal Requirements:

Reviewers' comments:

Reviewer's Responses to Questions

**Comments to the Author**

1. If the authors have adequately addressed your comments raised in a previous round of review and you feel that this manuscript is now acceptable for publication, you may indicate that here to bypass the “Comments to the Author” section, enter your conflict of interest statement in the “Confidential to Editor” section, and submit your "Accept" recommendation.

Reviewer #1: All comments have been addressed

2. Does this manuscript meet PLOS Global Public Health’s publication criteria ? Is the manuscript technically sound, and do the data support the conclusions? The manuscript must describe methodologically and ethically rigorous research with conclusions that are appropriately drawn based on the data presented.

Reviewer #1: Yes

3. Has the statistical analysis been performed appropriately and rigorously?

Reviewer #1: N/A

4. Have the authors made all data underlying the findings in their manuscript fully available (please refer to the Data Availability Statement at the start of the manuscript PDF file)?

Reviewer #1: Yes

5. Is the manuscript presented in an intelligible fashion and written in standard English?

Reviewer #1: Yes

6. Review Comments to the Author

Reviewer #1: Thank you for addressing all comments.

7. PLOS authors have the option to publish the peer review history of their article (what does this mean? ). If published, this will include your full peer review and any attached files.

**Do you want your identity to be public for this peer review?** For information about this choice, including consent withdrawal, please see our Privacy Policy .

Reviewer #1: **Yes: ** Liliana Abreu

---

## [Editor Report · Decision Letter 3]

21 Jan 2025

Barriers to implementation of domestic violence prevention policies and programs in northwestern Ethiopia: A qualitative implementation research

PGPH-D-24-00970R3

Dear Dr. Manu,

We are pleased to inform you that your manuscript 'Barriers to implementation of domestic violence prevention policies and programs in northwestern Ethiopia: A qualitative implementation research' has been provisionally accepted for publication in PLOS Global Public Health.

Best regards,

Julia Robinson

Executive Editor